# An Investigation of Compressive Creep Aging Behavior of Al-Cu-Li Alloy Pre-Treated by Compressive Plastic Deformation and Artificial Aging

**DOI:** 10.3390/ma16052054

**Published:** 2023-03-02

**Authors:** Jinqiu Liu, Fuqiang Guo, Kenji Matsuda, Tao Wang, Yong Zou

**Affiliations:** 1Key Laboratory for Liquid-Solid Structural Evolution and Processing of Materials, School of Materials Science and Engineering, Shandong University, Jinan 250061, China; 2Faculty of Sustainable Design, University of Toyama, Toyama 930-8555, Japan; 3Shandong Engineering & Technology Research Center for Modern Welding, Shandong University, Jinan 250061, China

**Keywords:** Al-Cu-Li alloy, KAM, compressive creep, pre-deformation, pre-aging

## Abstract

In this paper, the effects of compressive pre-deformation and successive pre-artificial aging on the compressive creep aging behavior and microstructure evolution of the Al-Cu-Li alloy have been studied. Severe hot deformation mainly occurs near the grain boundaries during the compressive creep initially, which steadily extends to the grain interior. After that, the T_1_ phases will obtain a low radius–thickness ratio. The secondary T_1_ phases in pre-deformed samples usually only nucleate on dislocation loops or Shockley incomplete dislocations induced by movable dislocations during creep, which are especially prevalent in low plastic pre-deformation. For all pre-deformed and pre-aged samples, two precipitation situations exist. When pre-deformation is low (3% and 6%), solute atoms (Cu and Li) can be consumed prematurely during pre-aging at 200 °C, with dispersed coherent Li-rich clusters in the matrix. Then, the pre-aged samples with low pre-deformation no longer have the ability to form secondary T_1_ phases in large quantities during subsequent creep. When dislocation entangles seriously to some extent, a large quantity of stacking faults, together with a “Suzuki atmosphere” containing Cu and Li, can provide the nucleation sites for the secondary T_1_ phase, even when pre-aged at 200 °C. The sample, pre-deformed by 9% and pre-aged at 200 °C, displays excellent dimensional stability during compressive creep because of the mutual reinforcement of entangled dislocations and pre-formed secondary T_1_ phases. In order to decrease the total creep strain, increasing the pre-deformation level is more effective than pre-aging.

## 1. Introduction

Al-Cu-Li alloys are extensively used for aerospace applications due to their high specific strength, good damage tolerance, and excellent property stability [1]. They can contain various precipitates such as θ′-Al_2_Cu, δ′-Al_3_Li, T_2_-Al_5_Li_3_Cu, T_B_-Al_7_Cu_4_Li, χ-Al_5_Cu_6_Li_2_, and T_1_-Al_2_CuLi [2,3,4]. The main strengthening precipitate in Al−Cu−Li alloys is a T_1_ phase, which forms as semicoherent platelets on {111}_Al_ planes and exhibits a hexagonal structure [5]. Plastic deformation prior to aging usually increases the density of fine strengthening precipitates through the introduction of dislocations acting as preferential heterogeneous matrix nucleation sites [6,7]. Kim et al. have found that the stretching treatment greatly accelerated the nucleation kinetics of the T_1_ phase at the expense of S′ phases in Al-Li-Cu-Mg alloys [8], where there exists another precipitation sequence: α-(SSSS, Supersaturated Solid Solution) → clusters → GPB zone + GPB II zone/S″ → S′/S (Al_2_CuMg). Gable et al. [9] have found that the size and quantity of the θ′ phase and the T_1_ phase of an Al-Cu-Li-X alloy vary with the pre-deformation degree in the same aging treatment.

Creep age forming (CAF), an advanced metal forming method for manufacturing large structures of aluminum alloys, can synchronize metal creep and age strengthening, greatly improving manufacturing efficiency. CAF has been applied widely to treat the upper wing skins of the Gulfstream IV/V, the B-1B long-range combat aircraft, and the Airbus A330/340/380 [10,11,12]. During CAF, the deformation and precipitates of aluminum alloys are constantly changing and interacting with each other [13,14]. During tensile CAF [15] for Al-Cu-Li alloys, the pre-deformed sample can be dominated by a larger number of T_1_ phases and a lower number of θ′ phases within the grain, at the cost of low enrichment of T_1_ at the GBs (grain boundaries). Another tensile CAF investigation of pre-stretched Al-Cu-Li alloys [16] reveals that appropriate pre-stretching can effectively inhibit the orientation precipitation effect [17,18,19,20,21] of T_1_ and promote the nucleation of T_1_. Additionally, for the uniaxial tensile creep test, there is a threshold stress close to the yield stress that has a limited promotion effect on T_1_ precipitates [22]. The external stress below and above the threshold stress promotes the precipitation of T_1_ precipitates by two different mechanisms: the promotion mechanism of lattice distortion produced by the elastic stress, and the promotion mechanism of the multiplication of dislocations produced by the plastic stress. The plastic external stress especially resulted in the best improvement to the strength and ductility of creep-aged alloys, synergistically. However, it is necessary to avoid using excessive plastic stress for creep aging because it may cause creep damage and degrade its mechanical properties. Ma et al. also suggest a change of the creep mechanism from diffusion creep to dislocation climbing with increasing stress [23]. However, studies about the uniaxial compressive creep behavior have not been systematically reported.

Therefore, in this paper, we focused on the investigation of compressive CAF by the introduction of compressive pre-deformation and pre-artificial aging and provided a basis for the application extension of Al-Cu-Li alloys by CAF. The characteristics of deformation behavior and microstructure evolution are also analyzed. Additionally, a proper pre-treatment to obtain good mechanical properties and steady creep deformation is discussed.

## 2. Experimental Procedure

The material investigated is a novel Al-Cu-Li cast alloy. Its actual chemical composition has been determined by ICP-AES (Inductively Coupled Plasma-Atomic Emission Spectroscopy, Model: Agilent 5110) and OES (Optical Emission Spectrometer, Model: SPECTROLAB), as listed in Table 1. The shape of samples cut from a cast billet has a square cross-section of 10 mm × 10 mm and a height of 15 mm, as illustrated in Figure 1a. The homogenization parameter is “460 °C/18 h → 525 °C/22 h → Air cooled” and solution treatment parameter is “530 °C/1.5 h → Water quenched” in a controllable resistance furnace. The pre-compression strains are 3%, 6%, and 9%, respectively, at a strain rate of 1 mm/min, as shown in Figure 2. The oil-bath resistance furnace is for additional pre-artificial aging, only for samples 3A#, 6A#, and 9A#. The product model for creep experiments is RDL-10, with the compression direction parallel to Z direction of the sample. The creep temperature should be controlled at 170 °C and a compressive stress of 200 MPa should be applied. The procedure of heat treatment process flow is shown through schematic diagram in Figure 1c, with the corresponding parameters listed in Table 2.

After creep, successive indentations were made at a distance of 0.5 mm apart along the central axes (Horizontal and vertical) on Z–Y section of each sample using a DHV-1000 digital Vickers hardness tester under a load of 200 g for 15 s. The hardness value for each sample was averaged across all the successive indentations. Electron backscattered diffraction (EBSD) data were acquired on Z–Y section via Oxford NordlysMax3 system and processed by Channel 5 software using JEOL-JSM-7800F. KAM (Kernel average misorientation) maps represent the average misorientation (within 5°) of each pixel with respect to its surrounding pixels. Misorientations over a certain value should be discarded in order to exclude the misorientations associated with discrete sub-grains and GBs. Furthermore, TEM samples were observed by JEOL-JEM-F200 with an accelerating voltage of 200 kV.

## 3. Results

### 3.1. Hardness Evolution

In Figure 3a, the hardness of sample 9A# is 128.37 HV_0.1_ before creep, and it reaches the peak (142.31 HV_0.1_) after an 8 h compression creep. The increment is 13.94 HV_0.1_. After that, sample 9A# becomes overaged. Sample 9# reaches the peak platform of hardness for an 8 h creep and obtains the increment of 31.37 HV_0.1_. Meanwhile, the difference in hardness between sample 9# and sample 9A# is 33.51 at the beginning of creep. For samples 3#, 6#, 3A#, and 6A#, the trend is reversed. Although the hardness of samples 3A# or 6A# is higher than samples 3# or 6# at the beginning of creep, the hardness of samples 3# or 6# quickly exceeds samples 3A# or 6A# and continues to increase until the creep finishes. The total hardness increment for samples 3# and 6# during a 12 h creep is 42.66 HV_0.1_ and 24.39 HV_0.1_, respectively. Meanwhile, these values are only 17.66 HV_0.1_ and 4.33 HV_0.1_ for samples 3A# and 6A#. In addition, the difference in hardness between samples 3# and 3A# or between samples 6# and 6A# is within 8 HV_0.1_ at the beginning of creep.

### 3.2. Local Misorientation

As seen in Figure 4a–c, the average local misorientation will increase due to pre-deformation. Furthermore, it seems dislocations are entangled seriously for sample 9#. Since the average local misorientation of solution-treated samples is 0.47°, dislocation proliferation starts to occur in sample 3# for its slightly higher value (0.58°). Pre-aging at 200 °C can promote recovery behavior. Here, the amplitude of the decrease in the average local misorientation between samples 3# and 3A#, 6# and 6A#, and 9# and 9A# is 0.12°, 0.10°, and 0.09°, respectively (Figure 4d–f). Since precipitation on dislocations during pre-artificial aging would retard recovery to a low level [24], the difference between samples 9# and 9A# before creep is smallest.

After the 12 h compressive creep test, the average local misorientation of all samples increases, as illustrated in Figure 5. Thus, the dynamic recovery and recrystallization have not consumed the dislocations induced by creep. Compared between Figure 4 and Figure 5, on the one hand, the amplitudes are 0.41°, 0.26°, and 0.22° for samples 3#, 6#, and 9#, respectively. On the other hand, the amplitude is slightly smaller for samples 3A#, 6A#, and 9A# (0.40°, 0.27°, and 0.02°, respectively).

### 3.3. Texture and Schmidt Factor

Seen from the average value of the Schmidt factor only, the difference of each sample is not significant, as shown in Figure 6. However, the Schmidt factor distribution of sample 6A#-12 h is more concentrated, because the crystal orientation in the compression direction is in a single direction, namely, in [110]_α_, as shown in Figure 7. This easily leads to continuous deformation along a single slip band, resulting in shape instability (Figure 3b). Since other samples have more texture components, multiple slips can enhance the stability of the shape of these samples [25].

### 3.4. EDS and Backscatter Image of SEM

As the strength of the GBs decreases during creep, the plastic deformation of the GBs occurs more easily. Normally, longer T_1_ phases will form on the GBs during artificial aging. However, the precipitates near the GBs are shorter than grain internal ones for sample 3# after an 8 h creep, as shown in Figure 8a. Therefore, violent collisions between dislocations and precipitates near the GBs occur. As shown by samples 6#-8h and 9#-8h in Figure 8b,c, the size and the distribution of precipitates seem similar, regardless of proximity to the GBs or in the grain interior. As shown by Figure 8d, some longer and thicker grain internal T_1_ phases exist after 8 h for sample 9A#.

### 3.5. Precipitate Identification and Micro-Structure Analysis by TEM

Figure 9a–d shows the selected area electron diffraction pattern (SADP), bright-field (BF), and dark-field (DF) TEM micrographs captured along the <110>_α_ zone axis for sample 3#-12h. The aperture used for DF is illustrated by a blue circle in Figure 9a. The main precipitates were T_1_, together with a small number of δ′/β′ and θ′ phases. Since the alloy has a low content of Mg, σ and S′/S phases are seldom seen. Further study reveals two kinds of T_1_ phases. One (Figure 9e), namely secondary T_1_, is formed dominantly during creep with few atomic layers thickness in <111>_α_ directions, which nucleated around the δ′/β′ phases (Figure 9f) for the convenient supply of Li atoms [26]. The other one (Figure 9g) originated from the insoluble initial coarse T_1_ phase after solution treatment, which will hinder dislocation movement with greater probability. A long thin T_1_ phase within the grain was also verified, as shown in Figure 9h,i.

As shown in Figure 10a, the diffraction spots of the δ′/β′ phases for sample 6#-12h were very weak. The initial T_1_ phases appeared in rough lines along the {111}_α_ habitual planes, both within the grain (Figure 10b) and near the GBs (Figure 10c), implying the initial T_1_ phases were twisted [27]. Sha et al. [28,29] have found that deformation has an influence on the orientation of the precipitate, alters the atomic configurations of the precipitate–matrix interface, and increases the misfit strain energy of the interface. Because of this lattice distortion, the interfacial energy between precipitate and matrix increases, leading to the achievement of high free energy and the re-dissolution of precipitates [30]. A part of an initial T_1_ phase changes its orientation gradually, and the rest suddenly changes to a fixed direction, which is indicated by blue lines in Figure 10d. Some initial T_1_ phases were fractured into several fragments, with the longitude direction deviating from <112>_α_ direction (Figure 10e). Since the T_1_ phase was twisted, the distance between (0001)_T1_ layers was shorter than 0.935 nm (Figure 10f). The distribution of the secondary T_1_ phases in sample 6#-12h was different from sample 3#-12h. As shown in Figure 10g,h, the density of the secondary T_1_ phases on the (11$\bar{1}$)_α_ plane was higher than that of the (1$\bar{1}$1)_α_ plane. Additionally, the structure of the tip of the plate-shaped T_1_ or θ′ phase was immature (Figure 10i–k).

For sample 9#-12h, the reflections at the 1/2 g {220}_α_ positions are caused by the θ′ and insoluble β′-Al_3_Zr phases, and the slight streaks along the <200>_α_ directions are related to θ′ phases viewed edge-on, as shown in Figure 11a. Compared among samples 3#-12h, 6#-12h, and 9#-12h, the contrast of the Al matrix in the BFs is uniform when the pre-deformation is large (Figure 11b,c). This means that either the δ′-Al_3_Li phases or solute atom clusters have been exhausted to supply the growth of the T_1_ phase. The β′-Al_3_Zr and θ′ phases remained in Figure 11d–f. Additionally, the dislocations tangle into a network structure near the initial coarse T_1_ phases (Figure 11g). It is obvious that the twisted initial T_1_ phase has partially transformed into another phase during creep (Figure 11h). The secondary T_1_ phases are verified in Figure 11i,j.

When pre-aged at 200 °C for 2 h, precipitation is quite different after 12 h creep. As shown in Figure 12a,b, the secondary T_1_ phase in sample 3A#-12h appear in small amounts. The β′-Al_3_Zr phase still exists with the δ′ phase wrapping around it, as shown in Figure 10c. The δ′/β′ composite appears as a fish-eye shape. The δ′ phase alone may no longer exist. The θ′ phase was hardly present. Instead, a large number of coherent clusters consisting of a few nanometers are dispersed in the matrix (Figure 10d) and display weak Z-contrast, especially when adjacent to the coarse initial T_1_ phase (Figure 10c). Another new phase has formed from the inside of the T_1_ phase, with the new phase and the Al matrix separated by the T_1_ phase a few atomic layers thick (Figure 10e,g,h). Combined with Figure 10b,e, the new phase also contains a large quantity of Cu atoms. Additionally, the secondary T_1_ phase appears as a thick plate with a small diameter (Figure 10f).

The precipitation constitution of sample 6A#-12h is similar to sample 3A#-12h (Figure 13). The secondary T_1_ phases lay on the (1$\bar{1}$1)_α_ plane rather than on the (11$\bar{1}$)_α_ plane. Additionally, some secondary T_1_ phases have been divided into sections of low radius–thickness ratio, with the tip or edge of the plates dissolved to some extent (Figure 13d).

The sample 9A#-12h is unique. It consists of the initial T_1_ phase, the secondary T_1_ phase, β′-Al_3_Zr, and atomic cluster. Although the existence of minor δ′ or θ′ phases cannot be ruled out, it does not influence the mechanical properties. Only the reflections of the T_1_ phases and the matrix are visible (Figure 14a). The cluster viewed under the [100]_α_ zone axis displays a low diffraction contrast in Figure 14c. The lattice distortion of the matrix near the interface of the cluster can be observed in Figure 14d. Compared with samples 3A#-12h and 6A#-12h, the density of the secondary T_1_ phase in sample 9A#-12h is quite high (Figure 14g,h). The reflections at the 1/2 g {220}_α_ positions in Figure 14e could be caused by an insoluble β′ phase which happened to be enclosed in the diffraction aperture. The mature and stable crystal structure of the secondary T_1_ phase is identified in Figure 14i.

## 4. Discussion

Normally, the T_1_ phase can nucleate on: (1) the dislocation and stacking fault; (2) the (sub-)grain boundary; (3) vacancies and octahedral holes (secondary defects formed by vacancies); (4) the Solute cluster; (5) the GP area; and (6) the Dispersion phase [31]. For pre-deformed samples only (3#, 6#, and 9#), the dislocation density continues to increase during creep, and the T_1_ is dominant. The deformation during creep near the GBs is more severe for sample 3#. Then, the initial T_1_ phases continue to crash into sections or dissolve to become spherical, which reduces the interfacial energy and makes the system stable. When dislocations become entangled around GBs, the dislocation density increases in the grain interior, where the initial T_1_ also becomes twisted or dissolved. The difference in hardness between samples 3#-12h and 9#-12h is very small; comparatively, the dislocation density of sample 9#-12h is higher. Thus, the contribution of precipitation strengthening of secondary T_1_ is great for sample 3#-12h. The movable dislocations in the grain interior, rather than the entangled ones around the GBs, tend to form dislocation loops or helical dislocations that benefit the nucleation of secondary T_1_ and the formation of the final closed quadrilateral structure. Additionally, due to the entangled dislocation network and faster solute diffusion, the transformation of the initial T_1_ phases easily occurred. The situation of sample 6#-12h are between samples 3#-12h and 9#-12h. On the one hand, the entangled dislocation network can accelerate the dissolution of initial T_1_ phases; on the other hand, the relatively low movable dislocation density is not beneficial for the formation of a compact structure consisting of four T1 phases, thereby widening phase spacing. This causes the abnormal phenomenon of the hardness of sample 6# fitting into the middle between samples 3# and 9# after just a few hours of creep to occur.

When it comes to samples 3A# and 6A#, the evolutionary path of microstructure has changed. Since the pre-aging temperature is relatively high (200 °C), vacancy clusters and dislocation loops cannot form easily [32,33,34], which impedes the formation of secondary T_1_. In addition, the solute atoms Cu and Li diffuse easily to join the initial coarse T_1_ or other stable phases, such as T_2_, δ-AlLi, θ, and Al-Cu-Fe in the GBs. The consequence is that solute atoms (especially Cu) prepared for the precipitation of secondary T_1_ are insufficient before creep aging. Although Li-vacancy clusters or δ′ phases can be continuously cut into smaller pieces by movable dislocation and dissolved steadily to provide Li atoms and the vacancy needed for the nucleation and growth of the T_1_ phase during creep, Li atoms and “free” vacancies would capture each other quickly rather than join the formation of T_1_ by long term diffusion. The evidence mentioned in Figure 12d and Figure 14d seems to suggest a new phase, whose size is between the Li-vacancy cluster and the GP-Li reported by Ma et al. [35]. Additionally, the precipitation of GP-T_1_ required a longer incubation time because of a higher activation energy due to the lower degree of Cu super-saturation in the solid solution. Finally, the hardness of samples 3A#-12h and 6A#-12h is quite low. For sample 3A#, the continuous plastic deformation in one direction can be coordinated or impeded by other slip systems due to multiple texture components. The reason for the size instability of sample 6A# may be the single texture component.

Nevertheless, sample 9A# still obtains sufficient strength even in an over-aged state after a 12 h creep. Hence, there must exist another mechanism for T_1_ nucleation. The dispersive “Suzuki atmosphere” by segregation of Cu and Li atoms in stacking faults on {111}α planes in the nucleation stage of T_1_ in a deformed Al-Cu-Li alloy has been reported [36,37,38]. The conjecture is, apart from GP-T_1_, vacancy–solute clusters, and dislocation loops (a/6<112>Shockley incomplete dislocation), the sample 9A# may have produced a large quantity of stacking faults before creep, making Li atoms and Cu atoms continuously aggregate to the nucleating T_1_ phase. The creep rate in the early creep stage is highly related to the pre-deformation level, rather than to pre-aging. The decreasing rate of the height is higher for samples 3# and 3A#, and lowest for samples 9# and 9A#, initially. An interesting result is that the nucleated T_1_ phase in sample 9A# before creep will not only hinder the recovery behavior greatly but also impede the creep deformation, especially in the grain interior, making the change in local misorientation between samples 9A#-0h and 9A#-12h just 0.02°, compared to 0.22° between samples 9#-0h and 9#-12h. Above all, the stacking faults and incomplete dislocation obtained from dislocation decomposition seem to play more important roles in the nucleation of the T_1_ phase during compressive creep.

## 5. Conclusions

In this paper, the effects of compressive pre-deformation and successive pre-artificial aging on compressive creep aging behavior and microstructure evolution of Al-Cu-Li alloys have been characterized through a comparative analysis of six different states (3#, 6#, 9#, 3A#, 6A#, and 9A#). Based on the results obtained, we can summarize the following conclusions:The pre-deformed samples (3#, 6#, and 9#) after creep mainly consist of a coarse initial T_1_ phase, a secondary T_1_ phase with minor β′/δ′ and θ′ phases. The dislocation loop or Shockley incomplete dislocation induced by movable dislocation may benefit the nucleation of the secondary T_1_ phase, especially with moderately low plastic deformation, as was shown in sample 3#. The dislocation density contributes more to the hardness of sample 9#-12h and accelerates the aging process to reach the peak-aged state earlier.For pre-deformed and pre-aged samples (3A#, 6A#, and 9A#), there exist two precipitation situations. If pre-deformation is not enough to produce the stacking fault, the solute atoms Cu and Li would be consumed prematurely before creep, rather than forming the secondary T_1_ phase during pre-aging at 200 °C, with dispersed coherent Li-rich clusters in the matrix. Then, the alloys (samples 3A# and 6A#) no longer have the ability to form a secondary T_1_ phase in any quantity during subsequent creep. In other words, when the dislocation entangles seriously to some extent, a large quantity of stacking faults together with a “Suzuki atmosphere” containing Cu and Li can provide the nucleation sites for secondary T_1_ again, even pre-aged at 200 °C.The creep rate in the early creep stage is highly related to pre-deformation level rather than pre-aging. The decreasing rate of the height is higher for samples 3# and 3A# initially, due to the high density of movable dislocation. The low radius–thickness ratio of the secondary T_1_ phase near the GBs indicates that severe compressive creep deformation first occurs near the GBs, which extends to the grain interior steadily. An inappropriate pre-deformation and pre-aged treatment (sample 6A#) would lead to a single texture component, which results in successive slip bands and an unstable region. Sample 9A#, with proper pre-treatment, displays excellent dimensional stability during compressive creep.

## Figures and Tables

**Figure 1 materials-16-02054-f001:**
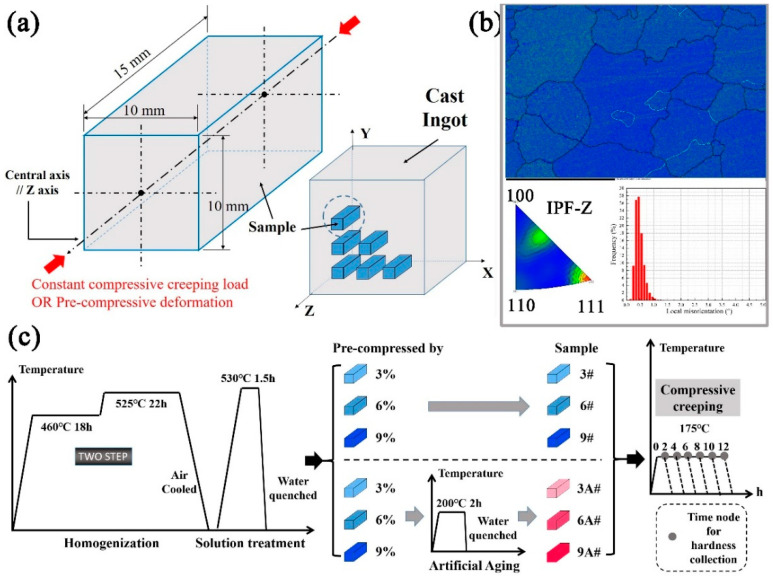
Schematic presentation: (**a**) dimensions for creep samples; (**b**) inverse pole figure in compression direction and KAM map, for solid-solution-treated sample; (**c**) heat treatment process flow.

**Figure 2 materials-16-02054-f002:**
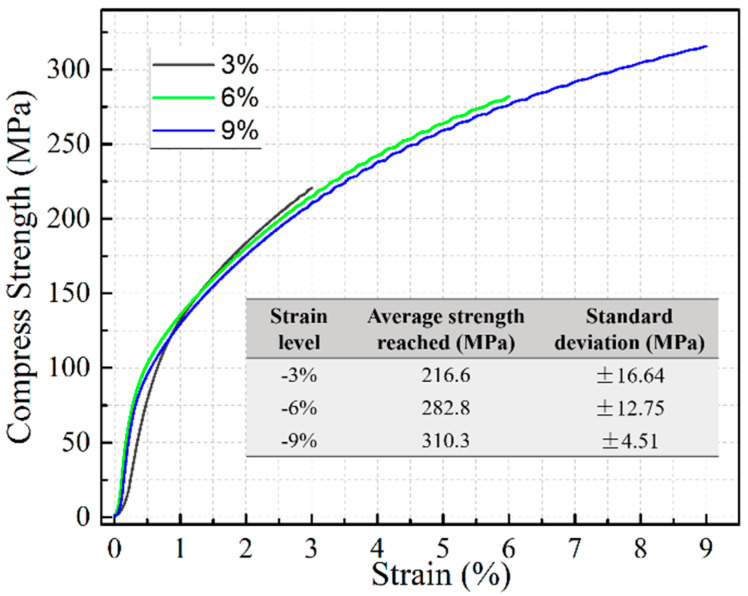
Pre-compressive deformation at different strain levels and corresponding strength reached.

**Figure 3 materials-16-02054-f003:**
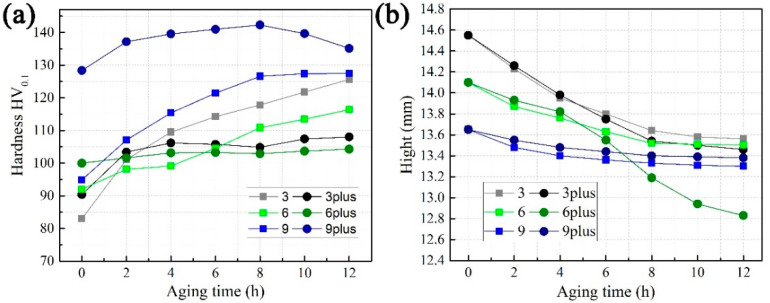
Variation of hardness (**a**) or height (**b**) as the function of compressive creep aging time.

**Figure 4 materials-16-02054-f004:**
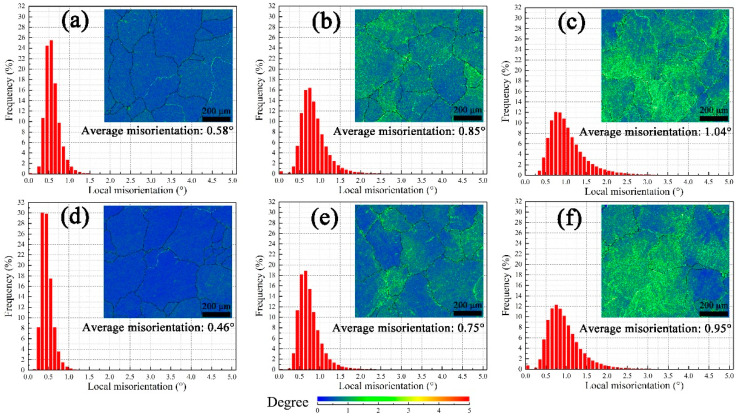
KAM maps before creep, pre-treatment: (**a**) 3#, (**b**) 6#, (**c**) 9#, (**d**) 3A#, (**e**) 6A#, (**f**) 9A#.

**Figure 5 materials-16-02054-f005:**
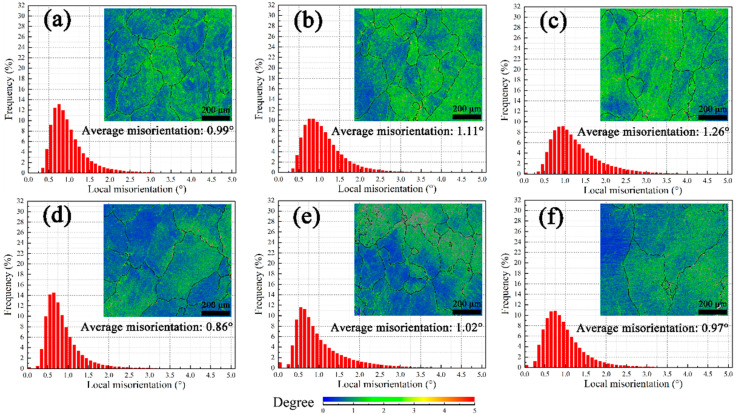
KAM maps of Al-Cu-Li alloy after 12 h creep, pre-treatment: (**a**) 3#, (**b**) 6#, (**c**) 9#, (**d**) 3A#, (**e**) 6A#, (**f**) 9A#.

**Figure 6 materials-16-02054-f006:**
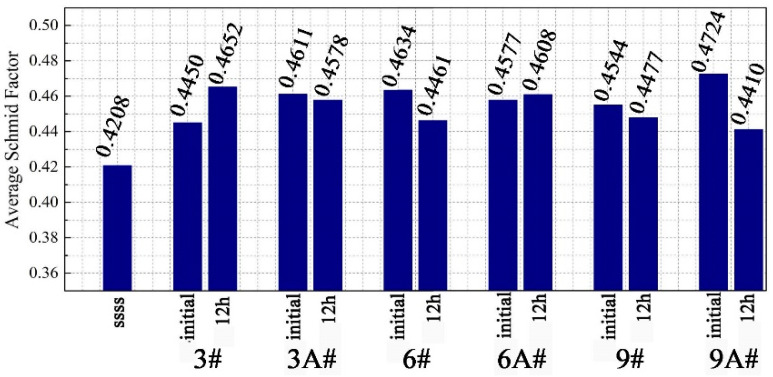
Schmidt factor evolution through compressive creep test.

**Figure 7 materials-16-02054-f007:**
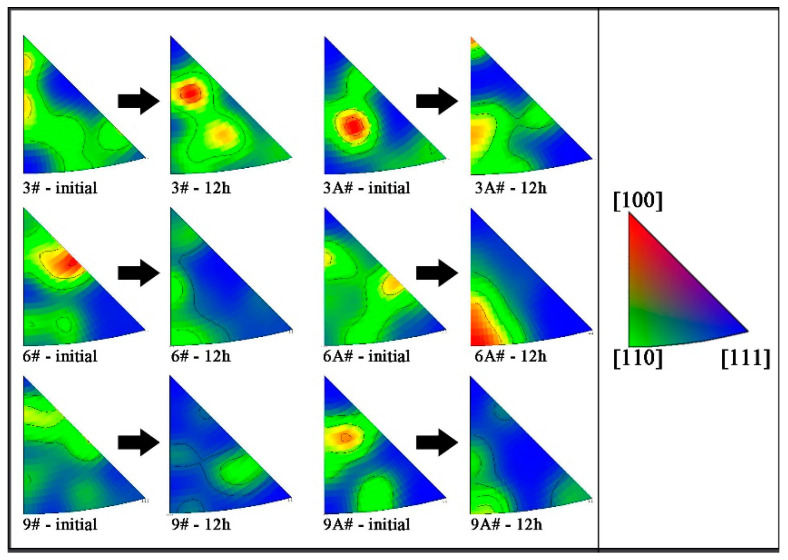
Inverse pole figure evolution in Z direction during creep, with EBSD data obtained in Z–Y cross-section.

**Figure 8 materials-16-02054-f008:**
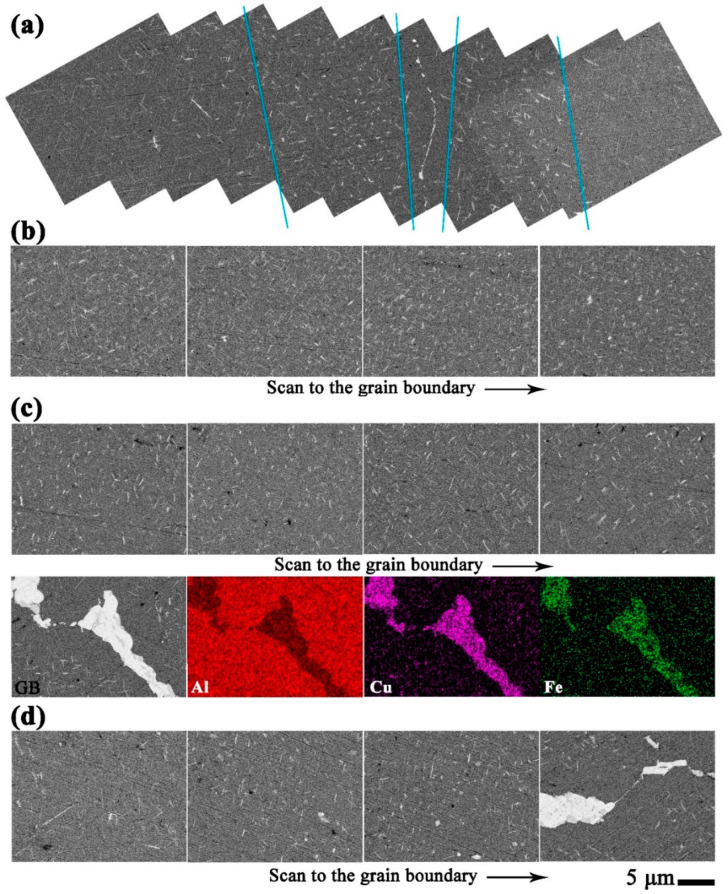
BSE images of the creep samples after 8 h: (**a**) 3#, (**b**) 6#, (**c**) 9#, and (**d**) 9A#.

**Figure 9 materials-16-02054-f009:**
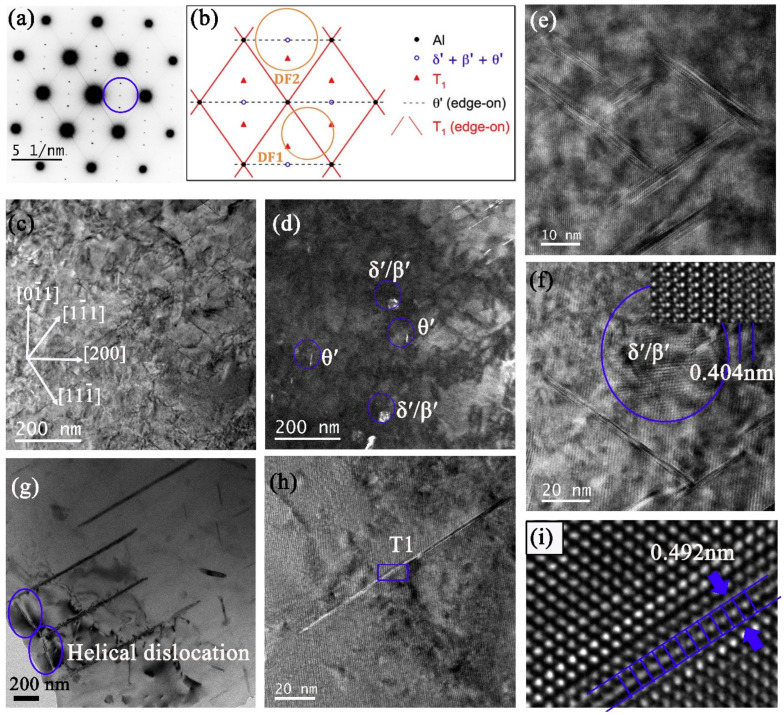
TEM analysis of 3# after 12 h creep. (**a**) SADP under [110]_α_ zone axis and (**b**) corresponding schematic SADP. (**c**) BF and (**d**) DF of (**c**) by selecting the diffraction spots in (**a**) using a blue circle. (**e**) Enlarged T_1_ phases forming a circle, (**f**) enlarged δ′/β′ phases with inset HRTEM, (**g**) helical dislocations together with thick T_1_ phases observed by multi-beam BF electron microscope technique, (**h**) a single long T_1_ phase, and (**i**) HRTEM of (**h**).

**Figure 10 materials-16-02054-f010:**
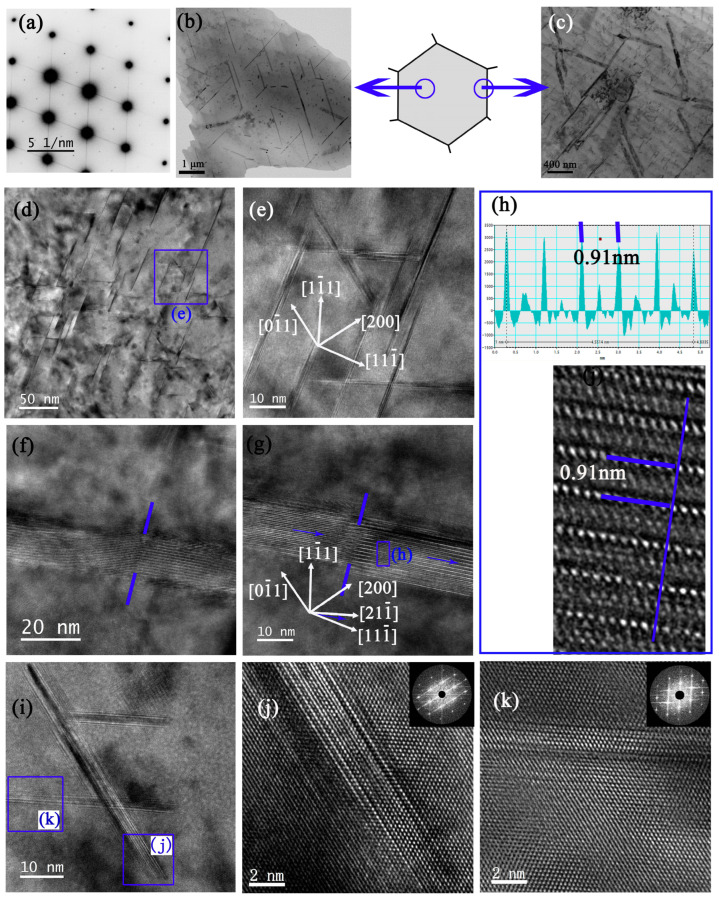
TEM analysis of 6# after 12 h creep. (**a**) SADP under [110]_α_. The BF of initial T_1_ phases within the grain (**b**) and near the grain boundary (**c**). (**d**,**e**): the arrangement of secondary T_1_ phase. (**f**–**h**): initial T_1_ phase. (**i**) Enlarged BF of T_1_ and θ′ phases, with corresponding HRTEMs of θ′ laid on (100)_α_ plane (**j**) and T_1_ on (1$\bar{1}$1)_α_ plane (**k**), respectively.

**Figure 11 materials-16-02054-f011:**
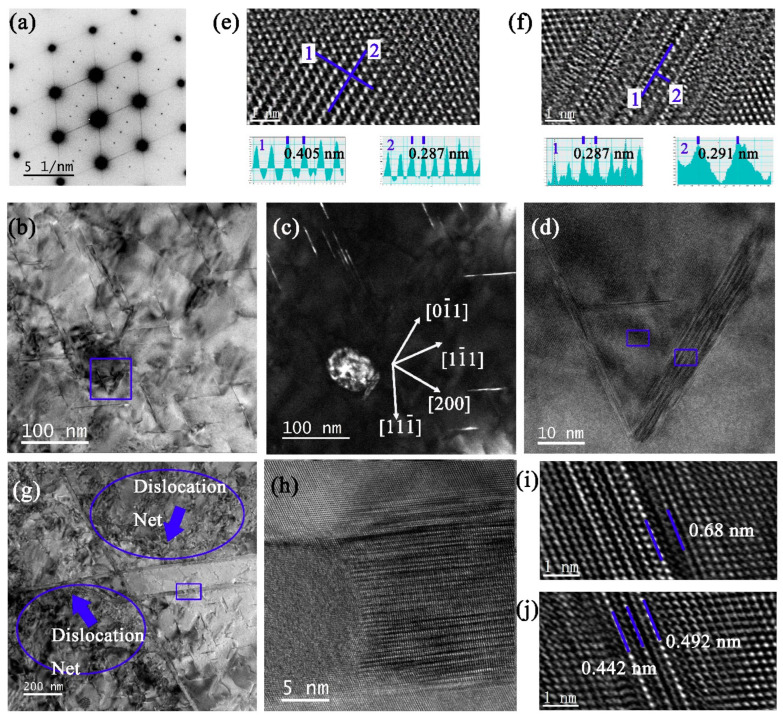
TEM analysis of 9# after 12 h creep. (**a**) SADP under [110]_α_. (**b**) BF and (**c**) DF of (**b**). (**d**) Enlarged BF selected by blue square box in (**b**), with HRTEM measured in (**e**,**f**). (**g**) BF with initial T_1_ phases and dislocation nets. (**h**) HRTEM of initial T_1_ phase by enlarging the blue box in (**g**). (**i**,**j**): two separate HRTEM images of secondary T_1_ phase.

**Figure 12 materials-16-02054-f012:**
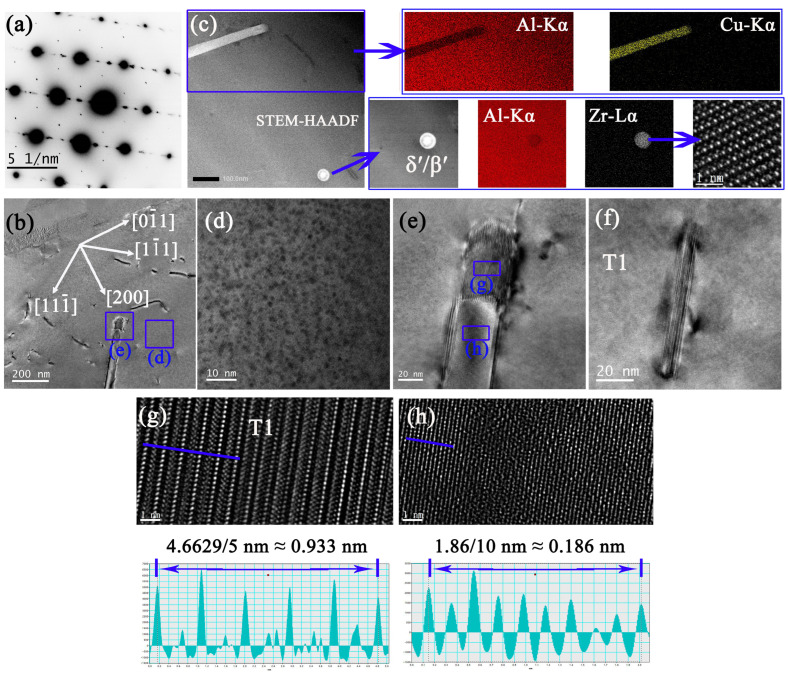
TEM analysis of 3A# after 12 h creep. (**a**) SADP under [110]_α_. (**b**) BF, (**c**) HAADF, and EDS mapping of the same area of (**b**) in STEM mode. (**d**,**e**): enlarged BF in (**b**). (**f**) BF of a single secondary T_1_ phase. (**g**,**h**) HRTEM images in (**e**).

**Figure 13 materials-16-02054-f013:**
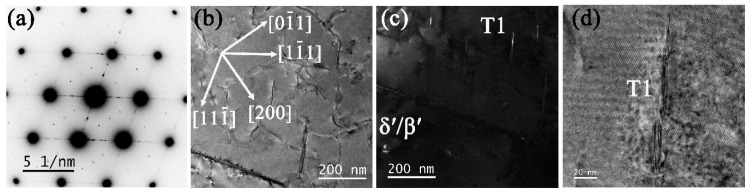
TEM analysis of 6A# after 12 h creep. (**a**) SADP under [110]_α_. (**b**) BF and (**c**) DF of (**b**). (**d**) BF of an enlarged secondary T_1_ phase.

**Figure 14 materials-16-02054-f014:**
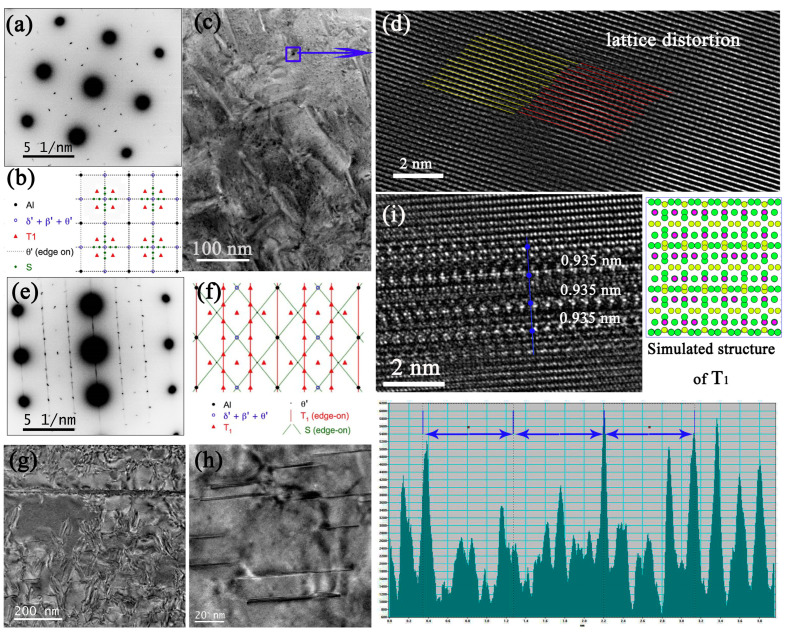
TEM analysis of 9A# after 12 h creep. (**a**) SADP under [100]_α_ and (**b**) corresponding schematic SADP. (**c**) BF viewed in [100] _α_ and (**d**) HRTEM of the blue box in (**c**). (**e**) SADP under [112]_α_ and (**f**) corresponding schematic SADP. (**g**,**h**) BF images viewed in [112]_α_. (**i**) HRTEM of a secondary T_1_ phase captured in [112]_α_.

**Table 1 materials-16-02054-t001:** Chemical composition of the investigated aluminum alloy (wt.%).

Cu	Li	Mn	Zr	Mg	Ti	Fe	Si	Co	Ce	Al
2.66	1.31	0.27	0.11	0.09	0.03	0.04	0.01	0.005	0.002	Bal.

**Table 2 materials-16-02054-t002:** Experimental parameters of creep samples.

Sample No.	Pre- Deformation	Pre- Artificial Aging	Compressive Creep Aging
3#	Strain: ε = −3%	-	170 °C/200 MPa
3A#	Strain: ε = −3%	200 °C/2 h	170 °C/200 MPa
6#	Strain: ε = −6%	-	170 °C/200 MPa
6A#	Strain: ε = −6%	200 °C/2 h	170 °C/200 MPa
9#	Strain: ε = −9%	-	170 °C/200 MPa
9A#	Strain: ε = −9%	200 °C/2 h	170 °C/200 MPa

## Data Availability

The data that support the findings of this study are available from the corresponding author, upon reasonable request.

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
