# Peer review of "An Investigation of Compressive Creep Aging Behavior of Al-Cu-Li Alloy Pre-Treated by Compressive Plastic Deformation and Artificial Aging"

_materials, 2023, doi:10.3390/ma16052054_

Round 1

Reviewer 1 Report

Abstract

pre-deformed samples (3#, 6# and 9#)

pre-deformed and pre-aged samples (3plus#, 6plus# and 9plus#)

Could you use some other marking the samples,  as exaple, 3,6,9 and  3A, , 6A,9A, respectivly.

2. Experimental Procedure

Please insert text before Figures at the beggining of chapters. It is unussual to start chapter with figues.

The same comment is for 3.1. Hardness evolution

Please, explain how did you chose this parameters: The creep temperature was controlled at 170 °C and applied a compressive stress of 200 MPa.

Refrences: please add some references from 2020-2023.

Author Response

Comments to the Author:

  1. Abstract: pre-deformed samples (3#, 6# and 9#); pre-deformed and pre-aged samples (3plus#, 6plus# and 9plus#). Could you use some other marking the samples, for example, 3, 6, 9 and 3A, 6A, 9A, respectively.

Answer: Thank you for the good advice.

We have uniformed sample numbers as 3#, 6#, 9#, 3A#, 6A# and 9A#, all through this manuscript. The places changed have been marked by yellow background.

  1. Experimental Procedure: Please insert text before Figures at the beggining of chapters. It is unusual to start chapter with figures. The same comment is for 3.1. Hardness evolution.

Answer: Thank you for the good advice.

We have put all the Figures after the corresponding text all through the manuscript.

  1. Please, explain how did you chose this parameters: The creep temperature was controlled at 170 °C and applied a compressive stress of 200 MPa.

 Answer: Thank you for the good advice.

Normally, there is a threshold stress, close to the yield stress, that has only a limited promotion effect on the precipitation of T1 precipitates. The external stress below and above the threshold stress promotes the precipitation of T1 precipitates by two different mechanisms. One is the promotion mechanism of lattice distortion produced by the elastic stress. Another is the promotion mechanism of multiplication of dislocations produced by the plastic stress. Both elastic and plastic external stress can synergistically improve the strength and ductility. Especially, the plastic external stress resulted in the best improvement to the strength and ductility of creep-aged alloys, synergistically. However, it is necessary to avoid using excessive plastic stress for the creep ageing because it may cause creep damage and degrade its mechanical properties. In this paper, the compressive strength is 216.6 MPa when the strain arrives at 3%, which is close to 200 MPa. Here, 200 MPa apparently provide a proper plastic external stress at the beginning of creep in this paper, which means larger than the yield stress at 170 ℃. In addition, 170 ℃ is a moderate temperature, below which the growth of δ′ and θ′ phases can be supported and above which T1 phase can be supported. Then, we think it is more convenient to study the essence of creep problem.

  1. Refrences: please add some references from 2020-2023.

Answer: Thank you for the good advice.

We have add references about the creep behavior of Al-Cu-Li alloy from 2020-2023 in Introduction, as followings:

Dandan Jiang, Ruibin Yang, Defa Wang, Zhongxia Liu. Effect of external stress on the microstructure and mechanical properties of creep-aged Al-Cu-Li-Ag alloy. Micron, 2021, 143: 103011.

Peipei Ma, Lihua Zhan, Chunhui Liu, Jianshi Yang, Kailiang Chen, Zhibin Huang. Strong stress-level dependence of creep-ageing behavior in Al–Cu–Li alloy. Materials Science & Engineering A, 2021, 802: 140381.
